# Genomic Organization of the B3-Domain Transcription Factor Family in Grapevine (*Vitis vinifera* L.) and Expression during Seed Development in Seedless and Seeded Cultivars

**DOI:** 10.3390/ijms20184553

**Published:** 2019-09-14

**Authors:** Bilal Ahmad, Songlin Zhang, Jin Yao, Mati Ur Rahman, Muhammad Hanif, Yanxun Zhu, Xiping Wang

**Affiliations:** 1State Key Laboratory of Crop Stress Biology in Arid Areas, College of Horticulture, Northwest A&F University, Yangling, Xianyang 712100, China; bajwa1999@nwafu.edu.cn (B.A.); zhangsonglin@nwafu.edu.cn (S.Z.); jin.yao@nwafu.edu.cn (J.Y.); mati@nwafu.edu.cn (M.U.R.); mhanif@nwafu.edu.cn (M.H.); yanxun.zhu@nwsuaf.edu.cn (Y.Z.); 2Key Laboratory of Horticultural Plant Biology and Germplasm Innovation in Northwest China, Ministry of Agriculture, Northwest A&F University, Yangling, Xianyang 712100, China

**Keywords:** B3 superfamily, transcription factor, ovule abortion, *Vitis vinifera*, expression analysis

## Abstract

Members of the plant-specific B3-domain transcription factor family have important and varied functions, especially with respect to vegetative and reproductive growth. Although B3 genes have been studied in many other plants, there is limited information on the genomic organization and expression of B3 genes in grapevine (*Vitis vinifera* L.). In this study, we identified 50 B3 genes in the grapevine genome and analyzed these genes in terms of chromosomal location and syntenic relationships, intron–exon organization, and promoter *cis-*element content. We also analyzed the presumed proteins in terms of domain structure and phylogenetic relationships. Based on the results, we classified these genes into five subfamilies. The syntenic relationships suggest that approximately half of the genes resulted from genome duplication, contributing to the expansion of the B3 family in grapevine. The analysis of *cis-*element composition suggested that most of these genes may function in response to hormones, light, and stress. We also analyzed expression of members of the B3 family in various structures of grapevine plants, including the seed during seed development. Many B3 genes were expressed preferentially in one or more structures of the developed plant, suggesting specific roles in growth and development. Furthermore, several of the genes were expressed differentially in early developing seeds from representative seeded and seedless cultivars, suggesting a role in seed development or abortion. The results of this study provide a foundation for functional analysis of B3 genes and new resources for future molecular breeding of grapevine.

## 1. Introduction

Deployment of specific regulatory genes at opportune times is key to the development and growth of a plant in a way optimized for its environment. Plants, like other higher organisms, use a cadre of DNA-binding transcriptional factors that act both alone and combinatorically to activate or repress regulatory genes. The plant-specific B3 DNA-binding domain was originally recognized in studies of the maize Vp1 (Viviparous-1) transcription factor, which carries out numerous developmental functions including repression of premature seed germination. The Vp1 protein is a member of a small protein family related by amino acid sequence that can be partitioned into five subfamilies: ABI3/VP1, HSI, RAV (Related to ABI3/VP1), ARF, and REM [1,2,3,4,5,6]; although, some researchers consider ABI3 and HSI as members of a single family designated as LAV (LEC2 (LEAFY COTYLEDON 2)/ABI3 (ABSCISIC ACID INSENSITIVE 3)/VAL (VP1/ABI3-LIKE)) [7]. The structure of the B3 domain comprises seven beta-strands (1–7) and two short alpha-helices (1 and 2). The beta strands form an open beta-barrel-like structure without closing the sheet, whereas the two short alpha-helices are located at either end of the barrel [8,9].

Several ARF and LAV family genes are transcriptionally regulated in response to auxin and abscisic acid [5,10,11]. ARF genes have been characterized in the context of many auxin-mediated physiological processes such as apical dominance, tropic responses, lateral root formation, vascular differentiation, embryo patterning, and shoot elongation [12,13]. LAV genes have been best characterized for their role in seed development and maturation [1,2,14,15,16]. RAV family members have not been as well characterized; however, some of them have been implicated in plant growth, development, and flowering time [17,18,19]. To date, the REM subfamily has been the least studied or characterized among all the B3 subfamilies [6]. However, it is known that the Arabidopsis REM gene VRN1 (VERNALIZATION 1) is involved in flowering [20,21].

Grapevine, including European (*Vitis vinifera* L.), American (*V. labrusca* L.), and Muscadine (*V. rotundifolia* L.), is one of the earliest fruit crops [22]. Grapes are cultivated worldwide and are consumed as fresh fruit, dried (raisins), juice, and wine [23,24]. Recently, the demand for seedless grapes is rising rapidly, fueling the development of seedless cultivars [25]. Seedlessness in grapes results from two distinct mechanisms: stenospermocarpy and parthenocarpy. Stenospermocarpy has great value as a research tool because the trait is heritable, is not strongly influenced by environmental factors, and because berry size is not greatly affected [26,27,28]. A large body of research has focused on potential mechanisms of seed abortion in stenospermocarpic grapes [29,30]. However, the key genes mediating this process have not been identified.

Recently, several genes encoding transcription factors have been implicated in seed development and abortion in grapevine. Overexpression of grapevine *VvCEB1*, encoding a helix–loop–helix transcription factor, affected embryo development and increased cell size [31]. Likewise, *VvAGL11*, encoding a MADS-domain transcription factor, was reported to have a role in stenospermocarpy [32]. Additional MADS genes, as well as some genes encoding homeodomain (HD) transcription factors, have been shown to be differentially expressed during development in seeded versus seedless grape cultivars [33,34]. In addition, *HD-ZIP* gene family members participate in the regulation of embryo abortion in grapes [35]. However, little is known about the role of B3 transcription factors during vegetative and reproductive development in grapevine. The important role of B3 family members as regulators of plant growth and developmental processes in other plants rationalizes a detailed bioinformatics and expression analysis of this gene family in this plant. In this study, we performed detailed bioinformatics analysis of B3 genes in grapevine, including gene structure and chromosomal locations, sequence homology, evolutionary duplication history, and cis-regulatory elements. We also analyzed expression of B3 genes during seed development in seeded and seedless grape cultivars. The results of this study will facilitate further studies of the mechanism of seed abortion in grapes.

## 2. Results

### 2.1. Genome-Wide Identification of B3 Genes in Grapevine

To identify B3 genes in the grapevine genome, we used a Hidden Markov Model (HMM) algorithm for the conserved B3 domain (PFAM 02362) in combination with the HMM search tool HMMer. A total of 61 putative B3 genes were revealed. All genes corresponding protein sequences were compiled and evaluated for the presence of an intact B3 domain using the Simple Modular Architecture Research Tool SMART (http://smart.emblheidelberg.de/) with default parameters [36]. The proteins having, incomplete B3 domains, DUF (domain of unknown function) or with non significant E-values were discarded. Finally, the integrity and accuracy of all the gene sequences were verified in the Grape Genome Database (12X). This approach led to the designation of 50 B3-domain-encoding genes. These genes were named according to their family name and their positions on the chromosomes [37,38,39]. Detailed information about genes including locus ID, accession number, chromosomal position, and length of coding sequence and open reading frame are given in Table 1.

### 2.2. Phylogenetic Analysis of Grapes, Arabidopsis, and Tomato

To depict the phylogenetic history of the B3 gene family in grapevine and to assist in their classification, a phylogenetic tree was constructed based on amino acid sequence alignment within the conserved B3 domain [8,9]. The analysis included 91, 50, and 82 B3-domain genes of Arabidopsis, grapes and tomato, respectively. According to the phylogenetic tree (Figure 1), we classified the proteins into five subfamilies: ARF, ABI3, HSI (VAL), RAV, and REM. The REM subfamily contained 49, 19 and 49 genes from Arabidopsis, grapevine and tomato respectively, whereas the LAV subfamily contained only 7 genes from grapevine and 6 from each Arabidopsis and tomato. This suggests that the REM family is more diverse and that the LAV subfamily has constant gene number in different species [39,40]. All 17 ARF subfamily genes could be further classified into 6 sibling pairs, whereas the remaining 5 VvARF were not matched. In the REM family, 19 genes were distributed into 4 sister pairs, whereas the remaining eleven genes were not matched with each other. In the RAV family, 7 genes were divided into 2 sister pairs, whereas the remaining three were not matched. However, in ABI3 and HSI subfamily each contained one sister pair of paralogous genes (Figure 2A). These results are consistent with previous findings that the REM subfamily is characterized by relative diversification and low bootstrap values [36]. The predicted length of proteins in the ARF, RAV, LAV, and REM subfamilies ranges from 486–1034, 228–380, 242–855, and 160–1085 amino acids, respectively.

### 2.3. Analysis of Gene Structure and Conserved Nucleotide Motifs

To gain insight into the phylogenetic relationships of grapevine B3 genes, an unrooted tree was constructed based on the B3 domain (Figure 2A). Consistent with the phylogeny including the Arabidopsis B3 genes, this supported the classification into five subfamilies: ARF (17 genes), REM (19 genes), VAL (4 genes), RAV (7 genes), and ABI3 (3 genes). The number of exons/introns varied in each subfamily, ranging from 2–15, 6–9, and 1–13 in ARF, ABI3, and REM families, respectively (Figure 2C). Members of the RAV and VAL subfamilies showed less variation in number of introns, suggesting that they are more conserved. In general, genes with similar intron numbers, positions and lengths corresponded to closely related siblings identified through the phylogeny, supporting the validity of the phylogeny. Furthermore, we generally observed similar motif distribution patterns within subfamilies (Appendix A). The highly conserved Motif 1 (B3 DNA binding domain; IPR003340) was found in almost all members of the grapevine B3 family. Motif 6, which corresponds to the auxin response factor domain (Auxin_resp; IPR010515), was found in all members of the ARF subfamily (Appendix A). These results are consistent with the conserved domains shown in Figure 2B. The RAV and ABI3 subfamilies were more conserved with respect to motif number and distribution. RAV and ABI3 have the same motifs (motifs 1, 9, and 15) and differ only in the distribution of the motifs. Three of the four members of the VAL subfamily exhibit relatively strong homology. The REM subfamily showed the most divergence in motif number and distribution. VvREM8, 14, and 15 contained only one motif, whereas VvREM4, 5, 6, and 7 contain three motifs (motifs 1, 9, and 18). The remainder of the REM representatives contained only two motifs. Moreover, members of the same clade of a phylogenetic tree shared similar motif organization with respect to either gene length or motif number. These results provided support to the reliability of phylogenetic analysis and Exon–intron distribution for classification.

### 2.4. Domain Architecture Analysis of Grapevine B3 Proteins

To gain further insight into the phylogenetic relationships among grapevine B3 domain genes, we analyzed their presumed protein products (open reading frame translations) for recognized peptide domains. In the ARF subfamily, three conserved domains (B3, AUX_RESP, and AUX_IAA) were identified (Figure 2B). All three domains were found in 12 of the 17 ARF proteins, whereas only two domains (B3 and AUX_RESP) were found in the remaining five (VvARF3, 4, 7, 8, and 9). We noted that three ARF proteins (VvARF2, 14, and 17) contained two AUX_IAA domains. Within the RAV subfamily, four RAV (VvRAV2, 5, 6, and 7) proteins contained AP2 and B3 domains, whereas the other three members had only one B3 domain. All three members of the ABI3 subfamily contained only a single B3 domain, whereas proteins included in the VAL subfamily contained a Zinc-finger domain along with the B3 domain. With one exception, VAL subfamily proteins contained both zf-CW and B3 domains. The domain architecture in REM proteins was more complex than that of the B3 subfamilies. Nine members of the REM subfamily contained multiple B3 domains. The highest number of B3 domains was noted in REM8, which contained six B3 domains.

### 2.5. Expansion Patterns of B3 Genes in Grape

According to annotation of the grapevine genome, B3 genes were broadly distributed and found on 16 of the 20 chromosomes. Chromosome 3 contained the highest number of B3 genes, eight, all of which belong to the REM subfamily. Two or more genes found on the same chromosome within a 200 kb region are likely to have resulted from tandem duplication [41]. According to these criteria, we observed twelve B3 genes (Appendix A) clustered by four tandem duplication events, on Chromosome 3 (two duplications), Chromosome 7 (one duplication), and Chromosome 18 (one duplication). Surprisingly, all of these genes belonged to the REM subfamily, indicating that REM has undergone more changes with the passage of time as compared with other B3 subfamilies. In addition to tandem duplication, we investigated eleven pairs of B3 genes apparently resulting from segmental duplication events (ARF13/ARF17, VAL1/VAL2, REM12/REM1, ARF4/ARF7, ARF5/ARF7, REM17/REM12, ARF2/ARF14, REM17/REM1, REM19/REM5, RAV3/RAV7, and RAV1/RAV6) (Figure 3, Appendix A), suggesting that both tandem and segmental duplication generated the grapevine B3 family. Interestingly, three genes REM 1, REM17, and ARF7, each were paired with two genes. REM1 was paired with REM 12 and REM 17, and ARF7 was paired with ARF4 and ARF7; whereas, in REM17, segmental duplication was observed with REM1 and REM12. In conclusion, 30 out of 50 (60%) of the B3 genes contributed to duplication events (segmental or tandem), which may provide a reference for the evolutionary relationship and functional potential of B3 genes.

### 2.6. Evolutionary Relationships among Grapevine, Tomato, and Arabidopsis B3 Genes

The function of B3 genes has been studied mainly in Arabidopsis. To further justify the origin, evolutionary history, and potential function of grapevine B3 genes, we examined genomic synteny of grape with tomato and Arabidopsis. A total of 16 pairs of syntenic relationships were identified between grapes and Arabidopsis, comprising 15 grapevine and 14 Arabidopsis genes (Appendix A, Figure 4). There were two pairs (AT2G30470-VvVAL1/VvVAL2 and AT3G19184-VvREM17/VvREM1) where a single Arabidopsis gene paired with more than one grapevine gene. Moreover, in grapevine, we also found one gene (REM17-AT3G19184/AT5G42700) that paired with two Arabidopsis genes. In the case of grapes and tomato, a total of 19 pairs of segmental duplications events were identified, comprising of 17 tomato and 19 grapevine genes (Appendix A). There were two pairs (Solyc08g013690.1-VvRAV3/VvRAV7 and Solyc02g079020.2-VvVAL3/VvVAL4) where a single tomato gene paired with more than one grapevine gene. In each syntenic block (grape and tomato or grape and Arabidopsis), both members belonged to the same subfamily and phylogenetic group. This suggests that they shared a common ancestor before evolution.

### 2.7. Expression Patterns of Grape B3 Genes during Seed Development

To identify B3 genes with potential function in seed development or abortion, we used semiquantitative PCR to examine expression of the B3 genes during progressive stages of seed development. Expression of members of the ARF family was previously reported by Wan et al. [42]. As shown in Figure 5A, most of the genes were expressed to similar levels between the seeded cultivar (“Red Globe”) and seedless cultivar (“Thompson Seedless”). However, some genes showed significantly different expression. For example, RAV3, RAV4, and REM2 were more highly expressed in Thompson Seedless than in Red Globe; in contrast, ABI3-1, ABI3-3, VAL1, and REM3 were more strongly expressed in Red Globe. To further support the results of semiquantitative PCR, nine genes determined to be differentially expressed between Red Globe and Thompson Seedless by semiquantitative PCR were analyzed by real-time, quantitative RT-PCR (Figure 5B). This supported that VvRAV3, VvRAV4, and VvREM2 were much more strongly expressed in Thompson Seedless as compared with Red Globe. The remaining six genes were more strongly expressed in Red Globe as compared to Thompson Seedless. For VvABI3-3 and VvVAL1, expression was almost undetectable in Thompson Seedless. These results suggest that these genes may have a role in seed development.

### 2.8. Developmental Regulation of Grapevine B3 Genes Outside of the Ovule

Analysis of the expression pattern of genes among various structures and organs is important for defining their functions. We used semiquantitative RT-PCR to analyze expression of the grapevine B3 genes in the root, stem, leaf, tendril and fruit of Thompson Seedless and Red Globe. We found that several of the genes, e.g., REM14, VAL2, VAL3, and RAV7, were expressed relatively ubiquitously, suggesting that they may have a general role in growth and development (Figure 6A). We noted that all the B3 genes were expressed in most of the grape structures. However, the level of expression of the B3 genes varied among the structures, or differed strikingly between Thompson Seedless and Red Globe. For example, ABI3-2 showed high expression levels in all structures of Red Globe, but a moderate level in roots, stems, leaves, and tendrils of Thompson Seedless and very low expression in fruits. In addition, ABI3-3 showed moderate expression levels in all structures of Thompson Seedless whereas no expression was detected in roots, leaves, or tendrils of Red Globe. To further elucidate the results of semiquantitative RT-PCR results, the nine genes identified as differentially expressed in developing seed (see above) were analyzed by real-time RT-PCR (Figure 6B). According to these results, ABI3-1, ABI3-3, RAV3, RAV4, REM1, and VAL3 were expressed more strongly in structures of Thompson Seedless as compared to Red Globe. The remaining three genes (VAL1, REM2, and REM3) were more strongly expressed in tissues of Red Globe as compared to Thompson Seedless.

### 2.9. Cis-Acting Elements and Gene Ontology (Go) Analysis of B3 Genes

To gain further insight into expression of the grapevine B3 genes, we carried out an in silico analysis of potential *cis-*elements conferring responsiveness to plant hormones (CGTCA, ERE, ABRE, P box, GARE, TGA-element, and AuxRR-core) within their promoters. *Cis-*elements related to disease resistance (TC-rich repeats and W-box) and stress response (TCA-motif, STRE, LTR) were also found in the promoter regions of 41 of the genes (Figure 7). Moreover, *cis-*acting elements involved in light responsiveness (G BOX, I BOX, BOX4, and GATA) and anaerobic respiration (ARE) were found in the promoter regions of almost all B3 genes.

We also carried out a gene ontology (GO) enrichment analysis of the grapevine B3 genes, considering biological process, molecular function, and cellular component, as previously described [37]. Prediction of biological process highlighted potential roles in transcription, fruit development, floral organ development, and response to abscisic acid (Figure 8). In addition, prediction of molecular functions of B3 proteins suggested that, as anticipated, most are involved in DNA binding. Analysis of cellular component suggested localization to various compartments including intracellular organelles and membrane-bound organelles.

## 3. Discussion

B3-domain transcription factors are known to have various roles in vegetative and reproductive development [36,43,44]. Members of the ARF and LAV B3 subfamilies have been well studied, whereas the RAV and REM subfamilies have remained less characterized [7]. Functional characterization of members of the B3 family has also implicated B3 domain genes in hormone signaling pathways including auxin, abscisic acid, brassinosteroid, and gibberellin. Genome-wide identification and expression analyses of the B3 gene family have been studied in several plants [40,45,46,47,48,49]. However, to our knowledge, this is the first comprehensive bioinformatics and expression analysis of B3 genes in grapevine.

In this study, we carried out comprehensive bioinformatics analysis of B3 genes in grapevine and compared their expression profiles between seeded and seedless cultivars during seed development and in various structures of the plant. We identified 50 B3 genes and classified these into five subfamilies based on phylogenetic analysis with Arabidopsis and tomato. As far as the number of genes in ARF and REM subfamilies in grapes is concerned, we observed that the previous trend, i.e., the relationship between ARF and REM gene numbers in one plant is generally, inversely proportional [39,40]. Similar to Arabidopsis and tomato, the REM subfamily comprises the largest group of B3 domain genes. Previously, 19 ARF genes were reported in grapevine [42]. In this study, we identified only 17, likely because we employed a more stringent E-value and other selection criteria. Within the ARF subfamily, twelve out of seventeen members contained three (B3, AUX_RESP and AUX_IAA) domains, with the remaining five (VvARF3, 4, 7, 8, and 9) containing only two domains (B3 and AUX_RESP). This suggests that the AUX_IAA domain has been less conserved than the B3 and AUX_RESP domains during the evolutionary process. The same observation have been reported in studies in other plants such as Arabidopsis, apple and papaya [12,45,47]. For the RAV subfamily our findings are concordant with previous results that most but not all proteins contain both B3 and AP2 domains [18]. According to the evolutionary relationships between the RAV proteins in Arabidopsis and grapevine, two RAV proteins (RAV3 and RAV4) are more closely related to NGA and NGAL in Arabidopsis, which do not have AP2 domains. In Arabidopsis, NGALs are involved in flower and seed development, whereas the NGA proteins are mainly involved in flower and leaf development [39,50]. Whereas, in tomato and cassava, the RAV transcription factor has showed resistant against bacterial blight [51,52]. Therefore, the RAV genes in grapevine may have important roles in flower, seed and leaf development as well as disease resistant. All members of the ABI3 subfamily contained only one B3 domain, whereas proteins included in the VAL subfamily contained a Zinc-finger domain as well. All members of the VAL subfamily contain a zf-CW domain, with one exception (VAL3). We speculate that loss of the zf-CW and AP2 domain occurred during evolution. The domain architecture in REM proteins is more complex than that seen in other subfamilies. Nine members of REM subfamily contain multiple B3 domains, with VvREM8 containing the highest number (six). These multiple domains may have evolved from one or more domain duplication events. During evolution, domain duplications can generate new genes along with different functions, thereby increasing functional diversity [9]. However, the function of genes with domain duplications generally is not easy to determine, because many of the proteins are functionally redundant.

The arrangement of exons and introns within a gene family can provide indications to support the evolutionary relationships. For B3 domain genes, in general, we found that genes within the same group tended to have the same number of introns, but we noted some exceptions. Evolutionary transition in exon number in genes within the same family plays an important role in the evolution of multiple gene families [53]. Duplicated genes are initially redundant, and consequently can adopt new functions [54]. The same phenomenon has been observed in other plants, such as Arabidopsis, canola, chick pea, hot pepper, rice, and tobacco [12,37,48,55,56].

Gene duplication is the primary source of evolution of genes and gene families [57]. Tandem and segmental duplications appear to be the main source of gene family expansion in grapevine, as the grapevine genome has not been duplicated during evolution [58]. We observed highly variable numbers of B3 domain genes on different chromosomes. Previously, it was reported that REM genes are clustered in the genome in at least 12 species, including Arabidopsis, maize, rice, sorghum, and tobacco [47]. Likewise, we identified eight REM subfamily genes only on chromosomes 3. This suggests that duplication of B3 genes probably occurred on chromosome 3 during evolution of the gene families. Interestingly, we also found that genes of the ARF and REM subfamilies are also clustered together on chromosome 8. The extensive gene duplication events may be responsible for the clustering. However, there is, at present, no practical experimental evidence to explain this observation [48]. According to our results in grapevine, 58% of the genes are products of tandem or segmental duplication events. Tandem duplications appear to have occurred in only eleven genes of the REM subfamily. Our results are in accordance with previous findings [36] that REM genes have evolved quickly and are functionally less conserved. In most of the plants, segmental duplication happens more frequently than tandem duplication because of polyploidy, which conserves numerous duplicated chromosomal blocks in their genome [57]. Consistent with this, we observed segmental duplication for 17 B3 genes. Additionally these duplicated genes clustered together in the same phylogenetic group. Two genes of the REM subfamily (1 and 12) exhibited both tandem and segmental duplication. In the REM subfamily, 73% (14/19) of genes appear to have experienced duplication events with the passage of time, whereas no duplication event was noted for ABI3 genes. These results indicated that ABI3 genes are highly conserved, whereas REM genes have been diversified. These results are consistent with previous findings that ABI3 genes are highly conserved among many plant species, including maize, Arabidopsis, rice, oat, wheat and tomato [59,60]. In this study, we found that the grapevine B3 gene pairs generated from tandem and segmental duplications showed conserved domains (Figure 2B), showing that these pairs persisted after the gene duplication event. However, there were three exceptions where duplicated genes showed a change in conserved domains—VvARF5/VvARF7 and VvRAV3/VvRAV7 (segmentally) and VvREM10/VvREM11 (tandemly). The main difference within these three pairs is the change in domain arrangements. In addition to gene duplication, rearrangements of domain structure can contribute to expansion of gene families and functional diversity [61]. Similar observations have been made in studies of chickpea, in which CaARF4/CaARF5 and CaARF21/CaARF22 are tandemly duplicated but structurally different [37]. These pairs might have been formed by ancient tandem duplication events.

Genomic synteny can provide clues to gene function, and can be an effective tool for gene family analyses where whole genome sequence is available [62,63]. We identified 16 gene pairs involved in segmental duplications between grapevine and Arabidopsis, and 19 pairs between grapes and tomato, suggesting they might have a common ancestor. A large number of synteny events suggests that many B3 genes arose before the divergence of the Arabidopsis, tomato and grapevine lineages. Although it is difficult to explain the evolutionary relationship between grapes, tomato and Arabidopsis based only on chromosomal syntenies, this analysis identified grapevine genes that share a common ancestor with their Arabidopsis and tomato B3 counterparts. Based on the reported functions of grapes orthologous genes in other species, we can predict possible functions of grapevine. For example, the At3g26790 (Fus3) orthologous of VvABI3-3, has role in seed development [15] and ectopic overexpression of SlABI3 resulted in the induction of seed-specific genes in vegetative tissues of tomato [64], while VvABI3-3 showed different levels of expression during seed development in seeded and seedless cultivars in this study (Figure 5B). These findings suggest that the grapes B3 genes endured a complicated evolutionary history. Taken together, these results enable further analyses of evolution and potential functions among these genes and will be helpful for a detailed study on the functions of homologous genes in various plant species.

The presence of *cis-*element motifs within promoter regions can provide clues for how genes might be regulated at the transcriptional level [63]. Our in silico analysis identified potential ABRE cis-elements, which participate in response to abscisic acid (ABA) in the promoter regions of 38 B3 genes. Interestingly, 17 of the 19 members of REM subfamily had both ABRE and ERE elements. This suggests that expression of these genes is influenced by ABA. In addition, GA-responsive elements (P box and GARE) and auxin-responsive elements (TGA-element and AuxRR-core) were found in some genes of the B3 family. Interestingly, all members of the RAV subfamily have GA-response elements, but only one member of RAV family contains an auxin-response element. This suggests that RAV family genes have a role in gibberellic acid signaling pathways. Presence of disease resistance and stress-responsiveness motifs suggests a role for B3 genes in disease resistance and stress signaling pathways. These results suggest that the regulation of B3 gene expression is complex and justifies the need for further research.

Gene expression patterns can provide important indication of gene function. In this study, we examined the expression of 25 B3 genes in five plant structures and six seed developmental stages in both seeded and seedless grape varieties. RAV3, RAV4 and REM2 were highly expressed in Thompson Seedless during various seed developmental stages. These genes might have a role in seed abortion of seedless grape cultivars. In contrast, ABI3-1, ABI3-3, and VAL1 showed high expression in the seeded cultivar Red Globe, suggesting that these three genes promote normal seed development. Our results are also supported by previous studies as AtLEC2 (Arabidopsis Leafy Cotyledon 2 gene) has been reported for its role in seed development and maturation [15,16], as well as its overexpression in cacao leaves changed the expression levels of several seed related genes [65].

Other B3 genes, including ABI3-2, VAL2, RAV3, REM5, REM6, and REM15, were expressed to similar levels during all stages of seed development in both cultivars, suggesting that these genes are functionally conserved in different grape cultivars and might have a general role in growth and development. For most of the genes, it was observed that, if a gene was highly expressed in fruit, it was also highly expressed in the seed. Other genes, such as ABI3-3, VAL3, and REM1, showed high expression levels in fruit of Thompson Seedless, but low expression during seed development. This same phenomenon was observed for the grapevine homeodomain family genes VvHB62, VvHB63, and VvHB55 [34]. In order to assess potential functional distinction of duplicated B3 genes, we compared their expression patterns in Red Globe and Thompson Seedless (Figure 6B). As projected, most of the duplicated genes were expressed in similar patterns between the seeded and seedless varieties. However, several examples of divergent expression patterns were also noticed. For example, VvRAV7 showed the same expression in all tissues and developmental stages of both cultivars, whereas VvRAV3 showed different expression profiles in both cultivars. The divergence in expression patterns between homologous and segmental duplication genes indicated that some of them might lose function or obtain new function after duplication in the evolutionary process.

## 4. Materials and Methods

### 4.1. Identification and Annotation of B3 Genes in Grapevine

To identify a complete list of B3 genes in the grapevine genome, annotated grapevine proteins were downloaded from three public databases: Grape Genome Database (http://www.genoscope.cns.fr), the National Centre for Biotechnology Information (NCBI; http://www.ncbi.nlm.nih.gov/), and the Grapevine Genome CRIBI Biotech website (http://genomes.cribi.unipd.it/). The Hidden Markov Model (HMM) profile of the B3 domain (PFAM 02362) downloaded from the Pfam database (http://pfam.xfam.org/) was used to survey all grapevine proteins in the 12X coverage assembly of the *V. vinifera* PN40024 genome to identify B3 genes [58]. The sequence integrity of the B3 domain (PFAM 02362) was assessed using SMART (http://smart.embl-heidelberg.de) with default parameters and genes with incomplete domains or with non-significant E-value were omitted from consideration [66]. All non redundant protein sequences with a conserved B3 domain were identified as grapevine B3 family members. Additionally, manual annotation was performed to resolve any discrepancy between incorrectly predicted genes and the actual chromosomal locations of involved genes in question.

### 4.2. Multiple Sequence Alignment and Phylogenetic Analysis

Multiple sequence alignments of 50 grapevine B3 sequences, 91 Arabidopsis B3 sequences and 82 tomato B3 sequences were performed using ClustalX 2.1 with default parameters [67]. The corresponding phylogenetic tree was constructed using the neighbor-joining (NJ) method and MEGA 6.0 software, using 1000 bootstrap replicates with the following parameters; “*p*-distance”, “Complete Deletion”, and gap setting [34]. Phylogeny among grapevine B3 sequences shown in Figure 2A was determined using MEGA 6 software by the neighbor-joining method with 1000 iterations and sequence alignments were performed using the ‘W’ approach.

### 4.3. Analyses of Exon–Intron Structure, Distribution of Conserved Motifs and Characteristic Domain Architecture

Exon–intron structures of the grapevine B3 genes were analyzed using aligned transcribed sequences and corresponding genomic sequences, and diagrams were created using the online Gene Structure Display Server 2.071 [62]. Conserved motifs and domains of proteins were analyzed using MEME 4.11.2 (http://meme-suite.org/tools/meme), searching up to 20 conserved motifs, and SMART (http://smart.embl-heidelberg.de). Protein domains were identified using the NCBI Conserved Domain Database (https://www.ncbi.nlm.nih.gov/Structure/cdd/cdd.shtml).

### 4.4. Synteny Analysis of Grape, Tomato and Arabidopsis B3 Genes

Tandem and segmental duplications of B3 genes in the grapevine genome were identified based on chromosomal locations. For synteny analysis, adjacent homologous grape B3 genes on a single chromosome without the presence of intervening genes was considered as a tandem duplication, whereas gene duplication events occurring on different chromosomes were defined as segmental duplications [68]. The list of grape B3 genes in duplicated genomic regions and a comparison of grape and Arabidopsis as well as grape and tomato genomes were retrieved from the Plant Genome Duplication Database [34]. Diagrams were generated using the program Circos version 0.63 (http://circos.ca/).

### 4.5. Analysis of Cis-Acting Elements and Gene Ontology

To examine the possible regulatory mechanisms of B3 genes, a 1.5-kb promoter region upstream of the start codon of each gene was retrieved from the Grape Genome Database (http://www.genoscope.cns.fr) and analyzed through the PlantCARE (http://bioinformatics.psb.ugent.be/webtools/plantcare/html/) online server. Gene ontology analysis of B3 protein sequences was acquired from Blast2GO (http://www.blast2go.com).

### 4.6. Plant Materials

This study used one seeded cultivar, Red Globe, and one seedless cultivar, Thompson Seedless (both *V. vinifera*). Plants were maintained in the grape germplasm resource orchard of Northwest A&F University, Yangling, China (34°20′ N 108°24′ E). The obtained plant structures were young roots, stems, leaves, tendrils, and fruits (42 days after full bloom, DAF). All samples were collected from plants under natural conditions. Developing seeds were dissected from fruits at 28, 31, 34, 37, 40, and 43 DAF. All samples were immediately frozen in liquid nitrogen and stored at −80 °C for RNA extraction and expression analysis.

### 4.7. RNA Extraction and Expression Analysis by PCR

Total RNA was extracted from samples using an EZNA Plant RNA Kit (R6827-01, OMEGA Biotek, Norcross, GA, USA), according to the manufacturer’s guidelines. First-strand cDNA was synthesized by reverse transcription of 500 ng total RNA using Prime Script RTase (Trans Gen Biotech, Beijing, China). After this, cDNA was diluted six-fold and preserved at −40 °C for further analysis. The grapevine *ACTIN1* gene (Genbank Accession NC_012010) and *EF1-α* gene (Genbank Accession NC_012012) were used as internal controls; *ACTIN1* was amplified with the oligonucleotide primers (5′-GAT TCT GGT GAT GGT GTG AGT-3′) and (5′-GAC AAT TTC CCG TTC AGC AGT-3′), and *EF1-α* was amplified using (5′-AGG AGG CAG CCA ACT TCA CC-3′) and (5′-CAA ACC CTG CAT CAC CAT TC-3′). Gene-specific primers were designed for selected B3 genes using Primer Premier 6.0 (Appendix A). The specificity of primers was checked in the NCBI (https://www.ncbi.nlm.nih.gov/) database, using the Primer-BLAST program. Semiquantitative RT-PCR assays were carried out in a volume of 20 μL per reaction containing 1 μL of cDNA template, 2 μL of gene-specific primers (1.0 μM), 7 μL sterile distilled water, and 10 μL PCR Master Mix (BIOSCI BIOTECH CO. LTD, Hangzhou, China). PCR conditions were 94 °C for 2 min, 35–38 cycles of 94 °C for 30 s, 51–63 °C for 30 s (depending on the specific gene), and 72 °C for 30 s, with a final extension of 72 °C for 7 min. In each case, 10 μL of the resulting product was resolved on a 1.5% (*w*/*v*) agarose gel and visualized using ethidium bromide and then imaged under ultraviolet light using GeneSnap software. Each assay was performed with three biological replicates. Semiquantitative RT-PCR expression data were visualized using GeneTools software. The mean expression value for each gene was determined in all tissues or seed developmental stages in all cultivars, and was then log2-transformed to generate heat maps using Multi Experiment Viewer software (Mev 4.8.1). Quantitative real-time PCR was performed for selected genes using SYBR Green (Trans Gen Biotech, Beijing, China) on an IQ5 real-time PCR machine (Bio-Rad, Hercules, CA, USA). *ACTIN1* was used as the internal reference gene and each reaction was performed with triplicate technical and biological repeats. The reaction mixture was 1 μL of cDNA template, 0.8 μL each primer (1.0 μM), 0.4 μL Rox reference dye, 7 μL sterile distilled water, and 10 μL of SYBR green. PCR was performed following the parameters 95 °C for 30 s, followed by 42 cycles of 95 °C for 5 s and 60 °C for 30 s. Relative expression levels were determined by the comparative CT method also referred to as 2^−ΔΔ*C*T^ method, where ΔΔ*C*T = [(C_T_ target gene–C_T_ control gene) Sample A–(C_T_ target gene–C_T_ control gene) Sample B] [69]. After this, the RT-PCR values were used to create graphs using Sigma Plot 10.0 [33].

## 5. Conclusions

Bioinformatics analyses of 50 B3 genes in grapevine revealed that gene duplication played a role in the expansion of the B3 family in grapes. It was also found that B3 genes from grapevine and Arabidopsis shared some common ancestor. Studies of phylogeny, gene structure, and conserved motifs of the B3 genes have further provided insights on the evolutionary history of the grapevine B3 genes. Taken together, these results underscore the potential importance for B3 genes grapevine growth and development, particularly for seed development. Therefore, our systematic study of these genes will help in identifying candidate genes for functional studies. It will also serve as a foundation and reference for future research regarding the molecular mechanisms controlling the seedlessness trait of grapes.

## Figures and Tables

**Figure 1 ijms-20-04553-f001:**
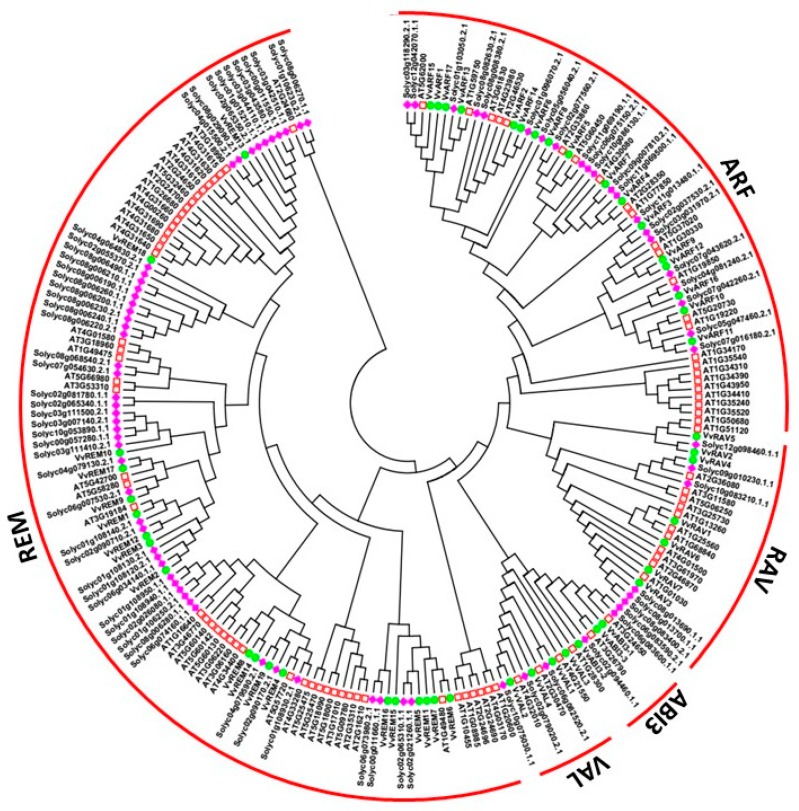
Phylogenetic analysis of B3 proteins from grapevine, tomato and Arabidopsis. Diamonds represent tomato protein, circles represent grapevine proteins, and squares represent Arabidopsis proteins.

**Figure 2 ijms-20-04553-f002:**
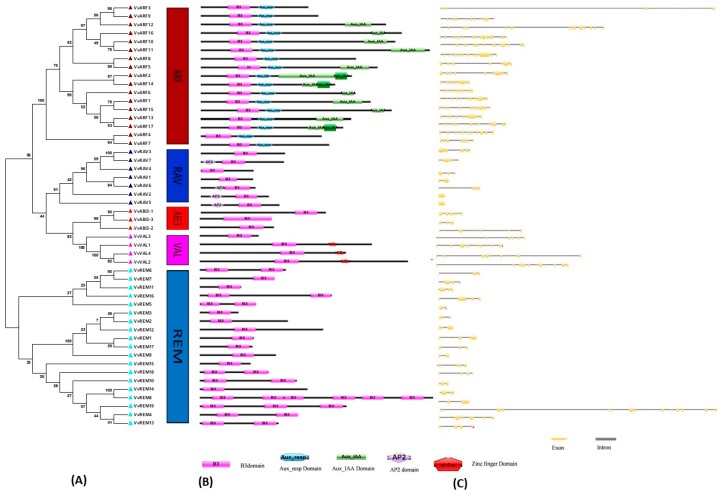
Structural analysis of grapevine B3 genes. (**A**) Phylogenetic analysis and classification. Different boxes are colored indicating different subfamilies. Numbers near the tree branches indicate bootstrap values. (**B**) Domain analysis. (**C**) Exon–intron structures. Exons are marked as yellow boxes, and introns are represented by black lines.

**Figure 3 ijms-20-04553-f003:**
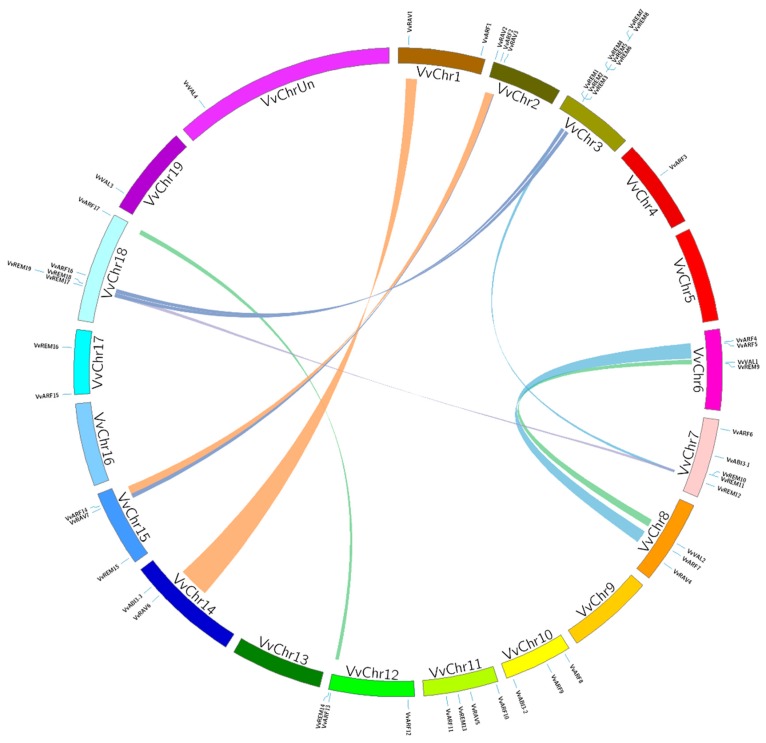
Synteny analysis and chromosomal distribution of grapevine B3 genes. Colored bars connecting two chromosomal regions denote syntenic regions; the corresponding genes on two chromosomes were regarded as segmental duplications. Chr: chromosomes.

**Figure 4 ijms-20-04553-f004:**
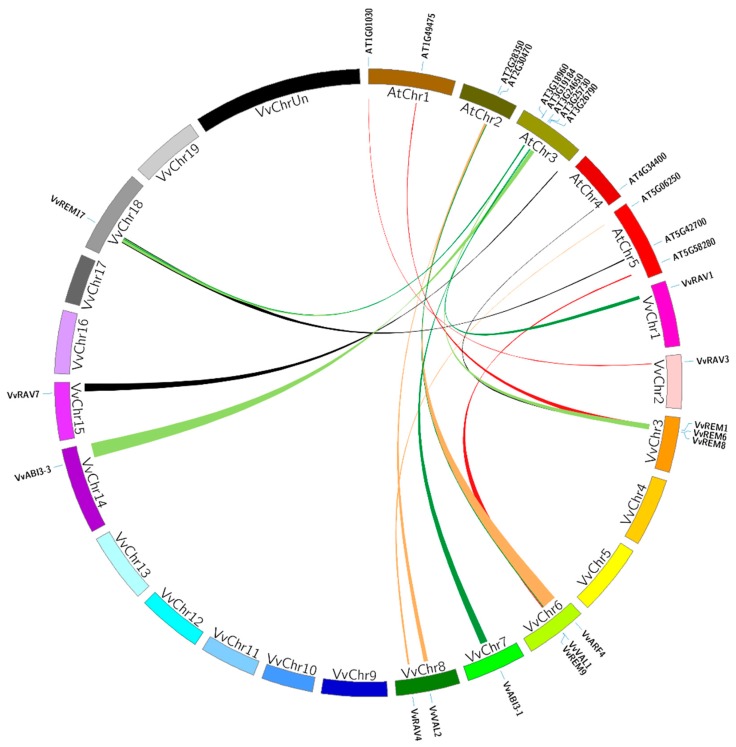
Synteny analysis of B3 genes between Arabidopsis and grapevine. Relative positions were depicted according to the grape and Arabidopsis chromosomes; colored lines represent syntenic regions.

**Figure 5 ijms-20-04553-f005:**
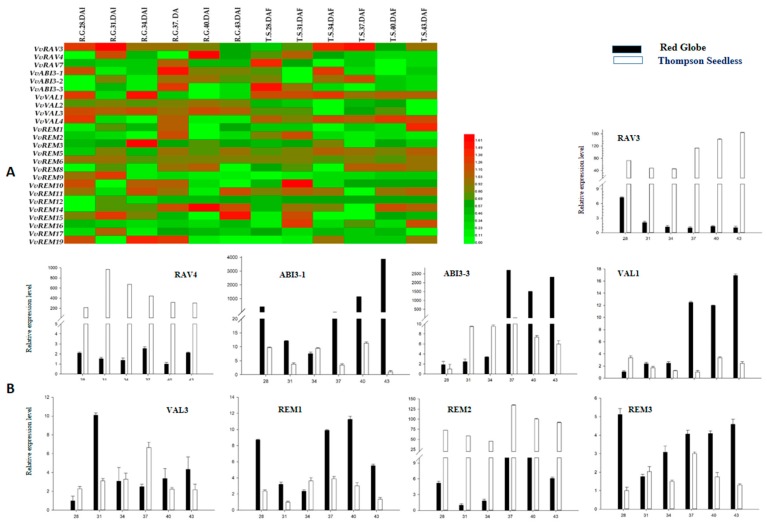
Expression profile analysis of B3 genes during progressive stages of seed development in seeded (Red Globe) and seedless (Thompson Seedless) cultivars. (**A**) Heat map of semiquantitative RT-PCR analysis. (**B**) Real-time PCR analysis. Numbers indicate the number of days after full bloom (DAF).

**Figure 6 ijms-20-04553-f006:**
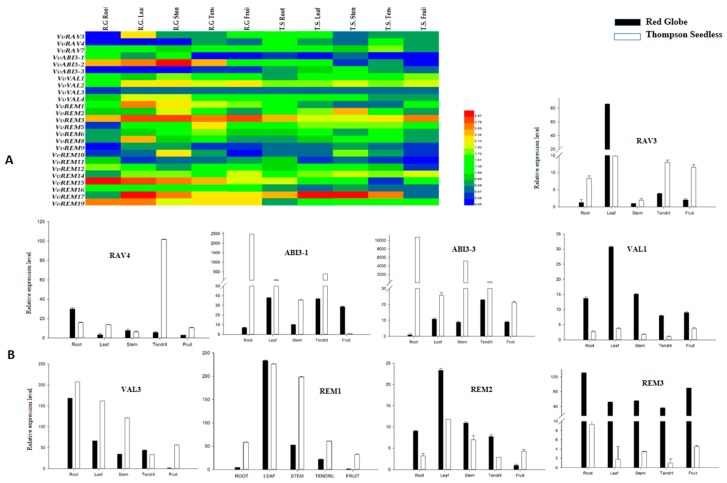
Expression analysis of grapevine B3 genes in various structures of seeded and seedless cultivars. (**A**) Heat map of Semiquantitative RT-PCR analysis. (**B**) Real-time PCR analysis.

**Figure 7 ijms-20-04553-f007:**
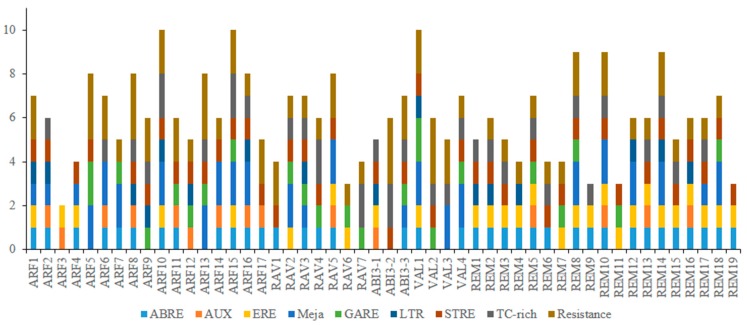
Predicted *cis-*elements in B3 gene promoters. The 1.5 kb sequences of 50 grape B3 genes were analyzed with the PlantCARE program.

**Figure 8 ijms-20-04553-f008:**
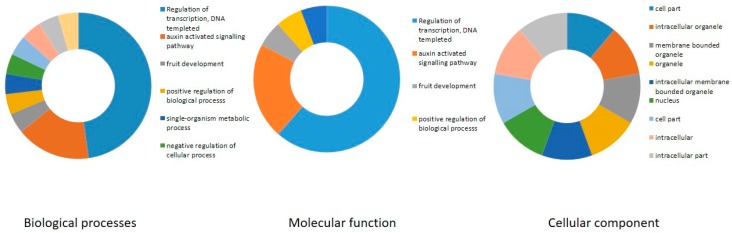
Gene ontology analysis of B3 proteins in three categories (Biological process, molecular function and cellular component) using Blast2Go program. Different colors which are indicated near the graphics represent distinct biological process, molecular function or cellular component.

**Table 1 ijms-20-04553-t001:** Characteristics of grapevine B3 genes.

Gene Locus ID	Gene ID	Accession No.	Chr. No.	Start Site	End Site	CDS (bp)	ORF (aa)
GSVIVT01004942001	*VvARF1*	CBI35669	1	21717511	21724801	2304	767
GSVIVT01019566001	*VvARF2*	CBI34510	2	1653587	1657804	2049	682
GSVIVT01035204001	*VvARF3*	CBI27334	4	10402895	10446692	1461	486
GSVIVT01025198001	*VvARF4*	CBI16322	6	3442998	3447976	1641	546
GSVIVT01025159001	*VvARF5*	CBI16287	6	3879687	3888448	2397	798
GSVIVT01011008001	*VvARF6*	CBI32272	7	2256656	2262296	2091	696
GSVIVT01025691001	*VvARF7*	CBI32737	8	12924236	12928912	1743	580
GSVIVT01021128001	*VvARF8*	CBI30623	10	1695978	1704736	2106	701
GSVIVT01021553001	*VvARF9*	CBI30950	10	6957710	6965185	1596	531
GSVIVT01015035001	*VvARF10*	CBI27770	11	629189	640325	2637	878
GSVIVT01032251001	*VvARF11*	CBI24055	11	13984914	13992724	3105	1034
GSVIVT01020805001	*VvARF12*	CBI22060	12	1745863	1766246	2511	836
GSVIVT01023149001	*VvARF13*	CBI29640	12	21939959	21950048	2037	678
GSVIVT01027166001	*VvARF14*	CBI40565	15	17300322	17304725	1824	607
GSVIVT01008639001	*VvARF15*	CBI15749	17	224846	230250	2589	862
GSVIVT01009865001	*VvARF16*	CBI19831	18	11920498	11929437	2724	907
GSVIVT01037136001	*VvARF17*	CBI16989	18	28749592	28756756	1932	643
GSVIVT01011947001	*VvRAV1*	CBI27062	1	2751620	2752908	687	228
VIT_202s0234g00085	*VvRAV2*	XP_019081631	2	110088	111017	930	309
GSVIVT01019699001	*VvRAV3*	CBI 34625	2	2549060	2551460	1143	380
GSVIVT01033902001	*VvRAV4*	CBI30344	8	16861285	16864429	720	239
VIT_211s0037g00010	*VvRAV5*	XP_002276492	11	7603337	7604681	1074	357
GSVIVT01036447001	*VvRAV6*	CBI16858	14	22050931	22055841	750	249
GSVIVT01027463001	*VvRAV7*	CBI38731	15	16539622	16542157	1128	375
GSVIVT01028540001	*VvABI3-1*	CBI37258	7	9431080	9434223	1695	564
GSVIVT01034419001	*VvABI3-2*	CBI35396	10	16556697	16566825	918	305
GSVIVT01033007001	*VvABI3-3*	CBI21292	14	25028976	25030714	864	287
GSVIVT01024559001	*VvVAL1*	CBI15813	6	8848747	8857227	2127	708
GSVIVT01030182001	*VvVAL2*	CBI18036	8	10751372	10767781	2571	856
GSVIVT01014432001	*VvVAL3*	CBI20373	19	3676945	3687513	729	242
GSVIVT01003212001	*VvVAL4*	CBI23327	Un	7958360	7975590	1806	601
GSVIVT01031761001	*VvREM1*	CBI32428	3	4170184	4173696	672	223
GSVIVT01031762001	*VvREM2*	CBI32429	3	4176729	4178437	1092	363
GSVIVT01031763001	*VvREM3*	CBI32430	3	4180639	4183390	483	160
VIT_203s0063g01415	*VvREM4*	XP_010647801	3	4859928	4864681	1218	405
GSVIVT01031850001	*VvREM 5*	CBI32502	3	4874233	4875367	696	231
GSVIVT01031852001	*VvREM 6*	CBI32503	3	4879760	4881735	1068	355
VIT_203s0063g01455	*VvREM 7*	XP_010647883	3	4886639	4887736	930	309
GSVIVT01031853001	*VvREM 8*	CBI32504	3	4887975	4921315	3255	1085
GSVIVT01024536001	*VvREM 9*	CBI15797	6	9077586	9081132	945	314
GSVIVT01005085001	*VvREM 10*	CBI40975	7	14390670	14392515	1200	399
GSVIVT01005087001	*VvREM 11*	CBI40976	7	14403658	14404557	516	171
GSVIVT01022143001	*VvREM 12*	CBI21437	7	16919585	16924427	1527	508
VIT_211s0037g01365	*VvREM 13*	XP_019078510	11	10865702	10866963	981	326
GSVIVT01023108001	*VvREM 14*	CBI29609	12	22478661	22484790	1332	443
GSVIVT01019293001	*VvREM 15*	CBI39994	15	1791086	1795242	633	210
GSVIVT01007552001	*VvREM 16*	CBI14875	17	12251718	12257309	1626	541
GSVIVT01009519001	*VvREM 17*	CBI19534	18	9025103	9026342	654	217
VIT_218s0001g11355	*VvREM 18*	XP_010664615	18	9665889	9667070	858	285
GSVIVT01009586001	*VvREM19*	CBI19591	18	9669874	9677462	1815	604

Abbreviations: Chr: chromosome; CDS: coding sequence; ORF: open reading frame; Un: unknown chromosome.

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
