# Peer review of "Genomic Organization of the B3-Domain Transcription Factor Family in Grapevine (Vitis vinifera L.) and Expression during Seed Development in Seedless and Seeded Cultivars"

_ijms, 2019, doi:10.3390/ijms20184553_

Round 1

Reviewer 1 Report

This manuscript is well-written and represents the good results for the Genomic organization of the B3-domain transcription factor family in grapevine.

Author Response

Reviewer 1: This manuscript is well-written and represents the good results for the Genomic organization of the B3-domain transcription factor family in grapevine.

Response: Thank you for your kind helps.

Reviewer 2 Report

In the present research, the authors performed detailed bioinformatics analysis of B3 genes in grapevine, including gene structure and chromosomal locations, sequence homology, evolutionary duplication history and cis-regulatory elements. In the previous version of the manuscript, the reviewer put comments about an approach to search B3 genes in a particular species; i.e., the genes were collected based on orthologous genes in Arabidopsis. The authors added tomato genes to search these genes in grapevine as well as Arabidopsis ones, and in result, the approach was improved. However, the number of the selected genes was not changed. The genes exemplified by the reviewer were not contained even in the present manuscript.

Considering this insufficiency, the grapevine genes collected by the authors may not contain all B3 genes in the species. Here, the reviewer proposes to use not only the genes from the Grape Genome Database by the Genoscope, but also those from the CRIBI (http://genomes.cribi.unipd.it/) and those from the Genome database of the National Center for Biotechnology Information (NCBI). Although all of these databases provide information on the grapevine genome, the genes or proteins are not necessarily shared with each other. The genes commented by the reviewer are contained in CRIBI and NCBI, but in Genoscope. The reviewer believes that genes in a gene family should be identified not only using a single genome, but using multiple genomes if such information on the genomes is not integrated.

Additionally, the authors should notice the reason to select tomato for revealing evolutionary traits of grapevine. It is difficult to consider that tomato is the most appropriate species to reveal the traits.

Author Response

Response to Reviewer 2

The reviewer proposes to use CRIBI (http://genomes.cribi.unipd.it/) and the Genome database of the National Center for Biotechnology Information (NCBI) to find out missing genes.

Response: Thank you for your advice. According to the provided guidelines we have improved our methodology (L401-L414). we have used three public databases: Grape Genome Database (http://www.genoscope.cns.fr.), the National Centre for Biotechnology Information (NCBI; http://www.ncbi.nlm.nih.gov/) and the Grapevine Genome CRIBI Biotech website (http://genomes.cribi.unipd.it/), to find complete list of B3 genes in grapevine. The Hidden Markov Model (HMM) profile of the B3 domain (PFAM 02362) downloaded from the Pfam database (http://pfam.xfam.org/) was used to survey all grapevine proteins in the 12X coverage assembly of the V. vinifera PN40024 genome to find B3 genes. A total of 61 putative B3 genes were revealed. All genes corresponding protein sequences were compiled and evaluated for the presence of an intact B3 domain using the Simple Modular Architecture Research Tool SMART (http://smart.emblheidelberg.de/) with default parameters. The proteins having, incomplete B3 domains, DUF (domain of unknown function) or with non-significant E-values were discarded. According to this criteria following mentioned 11 genes were not considered for further study.

GSVIVT01004261001, GSVIVT01001696001, GSVIVT01023582001, GSVIVT01031848001, GSVIVT01016266001, VIT_203s0063g00910, VIT_200s0184g00066, VIT_200s0184g00062, VIT_207s0161g00008, VIT_200s0184g00106, and   XP_010662797. Different genes were rejected due to different reasons for example, XP_010662797 just have DUF 313 (domain of unknown function).To check the integrity of B3 domain we have used three different tools

Simple Modular Architecture Research Tool SMART (http://smart.emblheidelberg.de/), NCBI Conserved Domain Database (https://www.ncbi.nlm.nih.gov/Structure/cdd/cdd.shtml) and Pfam database (http://pfam.xfam.org/).

According to our findings no B3 domain was found in XP_010662797. Whereas, GSVIVT01023582001, GSVIVT01031848001, and GSVIVT01016266001 were not considered due to incomplete B3 domain (Small only 20-30 aa) domain. The genes having E-values lower than threshold were also rejected. This approach led to the designation of 50 B3-domain encoding genes of grapevine for further study.

For Arabidopsis 118 B3 genes reported by Swaminathan et al. in 2008 were also checked according to above mentioned procedure and finally 91 genes were used to create phylogenetic tree between grapes and Arabidopsis. Interestingly, different scientists have mentioned different B3 Arabidopsis genes in their studies. For example 87, 92 and 95 B3 genes were reported respectively by Romanel et al. 2009, Peng and Weselake 2013 and Wang et al.2012. Finally, according to new number of genes we have made all relevant changes in article including table and pictures. In this way our research design, method and conclusions have been improved.

The authors should notice the reason to select tomato for revealing evolutionary traits of grapevine.

Response: Thank you for your advice. Tomato (Solanum lycopersicum) is mostly used as model plant to study fruit development in different plants. Although, grapes and tomato are not very close relatives but have many similarities. For example both are eudicots, having fleshy berry and parthenocarpic fruits. During the annotation genomes study of tomato and potato it was noticed that, the protein coding genes of tomato, potato, Arabidopsis, rice and grape were clustered into 23,208 gene groups, of which 8,615 are common to all five genomes, 1,727 are confined to eudicots (tomato, potato, grape and Arabidopsis), and 727 were confined to plants with fleshy (tomato, potato and grape) fruits (Nature volume485, pages635–641). Due to these reasons mostly tomato has been used to study phylogenetic history of grapes in alone or in combination with Arabidopsis. For example Y. Gao et al. in 2017 used tomato as a refrence plant to study Ubiquitin-Conjugating Enzyme (E2 Gene Family) in grapes. As well as Guo et al. in 2014 used tomato and Arabidopsis to study phylogeny of WRKY gene family in grapes. Many scientist have used tomato to study fruit development related genes of grapes. Ye et al. reported in 2018 that expression of grape ACS1 in tomato decreases ethylene and alters the balance between auxin and ethylene during shoot and root formation. Over-expression of a grape stilbene synthase gene in tomato induces parthenocarpy and causes abnormal pollen development (Ingrosso et al. 2011). As well as B3 family has been well studied in tomato and Arabidopsis as compared to close relative of grapes. Finally, we have used tomato just to support our results, most of the results are described on the basis of grape and Arabidopsis comparisons.

References:

Swaminathan, K.; Peterson, K.; Jack, T. The plant B3 superfamily. Trends Plant Sci. 2008, 13, 647–655.

Romanel, E.A.; Schrago, C.G.; Couñago, R.M.; Russo, C.A.; Alves-Ferreira, M. Evolution of the B3 DNA binding superfamily: New insights into REM family gene diversification. PLoS ONE 2009, 4, e5791.

Wang, Y.J.; Wang, S.X.; Deng, D.X.; Bian, Y.L.; Zhang, R.; Yin, Z.T. Systematic analysis of plant-specific B3 domain-containing proteins based on the genome resources of 11 sequenced species. Mol. Biol. Rep. 2012, 39, 16.

Peng, F.Y.; Weselake, R.J. Genome-wide identification and analysis of the B3 superfamily of transcription factors in Brassicaceae and major crop plants. TAG. Theoretical and applied genetics. 2013, 126, 1305–1319.

Guo, C. et al. Evolution and expression analysis of the grape (Vitis vinifera L.) WRKY gene family. J Exp Bot. 2014. 65, 1513–1528.

Gao, Y. Wang, H. Xin, S. Li, and Z. Liang, “Involvement of Ubiquitin-Conjugating Enzyme (E2 Gene Family) in Ripening Process and Response to Cold and Heat Stress of Vitis vinifera”, Scientific Reports, vol.7, no.1,2017. 

Ilaria Ingrosso, Stefania Bonsegna, Stefania De Domenico, Barbara Laddomada, Federica Blando, Angelo Santino and Giovanna Giovinazzo, Over-expression of a grape stilbene synthase gene in tomato induces parthenocarpy and causes abnormal pollen development, Plant Physiology and Biochemistry 49, 10, 2011.

Sato, S. et al. The tomato genome sequence provides insights into fleshy fruit evolution. Nature, https://doi.org/10.1038/nature11119 10.1038/nature11119.

Round 2

Reviewer 2 Report

The authors added the sufficient revision to the reviewer's comments.

This manuscript is a resubmission of an earlier submission. The following is a list of the peer review reports and author responses from that submission.

Round 1

Reviewer 1 Report

This manuscript compares the genomic organization of the B3-domain transcription factor family of grapevine with that of Arabidopsis and shows the results of gene expression compared with seed development in two grapevine cultivars. However, in order for the manuscript to be published on MDPI, several things have to be corrected and edited.

1. How many times did the semi-quantitativ PCR experiment repeat? At least three times should be repeated experiments. The results of agarose gel electrophoresis of semi-quantitative PCR experiments should be presented as a supplementary data.

2. How was the quantitative real-time PCR experiment converted to a relative expression level through IQ5 software? What is the relationship between the relative expression level and the Ct value? The original graph of the real-time PCR experiment should also be presented as a supplementary data.

3. In Table 1, Chromosome -> Chromosome No., Start -> Start site, End -> End site, Chr1 -> 1, and so on

4. 'cis' should be italicized and please unify 'cis-element' notation.(Lines 241, 242, 248, 352, etc)

5. 'Semi-' -> 'semi' (Line 215)

6. 'material' -> 'materials' (Line 424)

7. 'reverse transcription' -> 'expression analysis by PCR'

8. In Figure 2 legend, (a), (b), and (c) should be changed to (A), (B), and (C), repectively.

9. In Figure 2, subfamily A-D must be replaced with another character to distinguish it from panel A-C.

10. '(Z. Y.)' should be in the same style as other text characters. (Line13)

Reviewer 2 Report

This manuscript is about bioinformatics analysis of B3 genes in grapevine, including gene structure and chromosomal locations, sequence homology, evolutionary duplication history and cis-regulatory elements. It is helpful for researches and cultivation of plants to curate such gene families in useful plants like grapevine. Therefore, such curation should be prudently performed. The tools and software to be used for the present study are agreeable. The approach to B3-domain genes in grapevine is not acceptable for detecting all members of the family. The authors obtained such genes by using Arabidopsis genes included in the family as queries. If grapevine is a species closely related to Arabidopsis, this approach is appropriate. However, they are not relative organisms. Some of the grapevine genes that contain B3 domain according to the Reference Sequence and Conserved Domain Database of National Center for Biotechnology Information (e.g., XP_010647801 and XP_010662797) may not be detectable through the approach because of their low similarities with Arabidopsis ones.

Furthermore, the authors discuss the evolutionary traits of the grapevine genes in comparison with Arabidopsis ones in Conclusion. Repeatedly, grapevine is not closely relative to Arabidopsis. For discussing the traits, genes of more plants in the family should be compared with those of grapevine.